# Genome-wide analysis clarifies the population genetic structure of wild gilthead sea bream (*Sparus aurata*)

**Francesco Maroso**[1☉¤*], **Konstantinos Gkagkavouzis**[2☉], **Sabina De Innocentiis**[3], **Jasmien Hillen**[4], **Fernanda do Prado**[5], **Nikoleta Karaiskou**[2], **John Bernard Taggart**[6], **Adrian Carr**[7], **Einar Nielsen**[8], **Alexandros Triantafyllidis**[2], **Luca Bargelloni**[1], **the Aquatrace Consortium**[¶]

1 Department of Comparative Biomedicine and Food Science, University of Padua, Legnaro (PD), Italy, 2 Department of Genetics, Development and Molecular Biology, School of Biology, Aristotle University of Thessaloniki, Thessaloniki, Macedonia, Greece, 3 ISPRA, Institute for Environmental Protection and Research, Roma, Italy, 4 Laboratory of Biodiversity and Evolutionary Genomics, University of Leuven, Leuven, Belgium, 5 Department of Biological Sciences, São Paulo State University, Bauru, Brazil, 6 Institute of Aquaculture, University of Stirling, Stirling, United Kingdom, 7 Fios Genomics Ltd, Edinburgh, United Kingdom, 8 Section for Population Ecology and Genetics, National Institute of Aquatic Resources, Technical University of Denmark, Silkeborg, Denmark

☉ These authors contributed equally to this work.
¤ Current address: Department of Life Science and Biotecnology, University of Ferrara, Ferrara, Italy
¶ Membership of the Aquatrace Consortium is provided in the Acknowledgments.
* francesco.maroso@gmail.com

**Data Availability Statement:** Demultiplexed raw reads files are available from the NCBI's SRA database (BioProject accession number PRJNA643702).

## Abstract

Gilthead sea bream is an important target for both recreational and commercial fishing in Europe, where it is also one of the most important cultured fish. Its distribution ranges from the Mediterranean to the African and European coasts of the North-East Atlantic. Until now, the population genetic structure of this species in the wild has largely been studied using microsatellite DNA markers, with minimal genetic differentiation being detected. In this geographically widespread study, 958 wild gilthead sea bream from 23 locations within the Mediterranean Sea and Atlantic Ocean were genotyped at 1159 genome-wide SNP markers by RAD sequencing. Outlier analyses identified 18 loci potentially under selection. Neutral marker analyses identified weak subdivision into three genetic clusters: Atlantic, West, and East Mediterranean. The latter group could be further subdivided into an Ionian/Adriatic and an Aegean group using the outlier markers alone. Seascape analysis suggested that this differentiation was mainly due to difference in salinity, this being also supported by preliminary genomic functional analysis. These results are of fundamental importance for the development of proper management of this species in the wild and are a first step toward the study of the potential genetic impact of the sea bream aquaculture industry.

## Introduction

Seen from a "terrestrial" perspective, the marine environment looks like a vast space with no barriers to limit the movements of the organism that inhabit it. This feature has a consequence

**Funding:** The project is funded by the 7th Framework Programme for research (FP7) under "Knowledge-Based Bio-Economy - KBBE", Theme 2: "Food, Agriculture and Fisheries, and Biotechnologies" Project identifier: FP7-KBBE-2012-6-single-stage Grant agreement no.: 311920. Also, Dr Gkagkavouzis was funded by a PhD scholarship of Alexander S. Onassis Public Benefit Foundation (GR). Fios Genomics Ltd. provided support in the form of salaries for authors AC, but did not have any additional role in the study design, data collection and analysis, decision to publish, or preparation of the manuscript. The specific role of this author is articulated in the 'author contributions' section.

**Competing interests:** The authors and the commercial company involved declare that they have no competing financial interests or personal relationships that could have influenced the work reported in this paper. The commercial affiliation with Fios Genomics Ltd. does not alter our adherence to PLOS ONE policies on sharing data and materials.

for population geneticists: compared to studies of terrestrial animals, exploring population genetic structure in marine species can be much more challenging. This is particularly true for fish, which are among the most motile marine organisms, often throughout their entire life cycle. New genomic approaches such as Next Generation Sequencing (NGS) and Restriction-site Associated DNA (RAD) have already proven to be effective in the detection of hidden population structure and in better defining indistinct genetic subdivision and differentiation in fish [1–6]. The greater number of informative markers that that are detectable by these new approaches also increases the chances of detecting genomic regions under selective pressure [2, 7, 8] allowing the investigation of demographic *vs* environmental causes of genetic differentiation among populations. Such molecular markers can also be exploited for the geographical traceability of wild samples, applicable in actions against illegal, unreported and unregulated (IUU) fishing as well addressing the increased interest of customers regarding food origin. Furthermore, they also enable detailed comparison of wild and farmed populations to predict the potential impact of aquaculture on natural populations, a recommended practice since the early '90s [9–14].

Gilthead sea bream is a protandric hermaphrodite, demersal species living in warm coastal and euryhaline waters of the Mediterranean Sea and North-East Atlantic Ocean [15]. As a highly prized culinary fish sea bream is an important target for both commercial and recreational fishing. Capture fisheries have provided almost constant yields since the '60s (around 8000 tons per year) with aquaculture production becoming increasingly important from the early '90s, reaching 186,000 tons in 2016 [16]. Following the expansion of marine cage culture, concerns have arisen regarding the potential effects of farm escapes on the natural populations [14, 17].

Current knowledge of gilthead sea bream genetics is scarce and fragmented for wild populations. Previous studies of the natural genetic structuring along its distribution range has not provided a consistent scenario, and while some surveys report an absence of genetic differentiation among basins [18], others report subtle genetic structure or population subdivision. These include evidence of differentiation between different Northwest Atlantic areas and the Mediterranean [19] but also at within-basin geographical scale, including along the European Atlantic coast from Portugal to Ireland [20] and even at finer scale within the Tyrrhenian Sea and the Adriatic Sea [21, 22].

Furthermore, it is likely that aquaculture practices, including those involving other species (e.g. tunas or shellfish), have an effect on shaping the population genetic structure of the species [23]. Most studies to date have been based on markers (e.g. microsatellites, mitochondrial DNA) which are nowadays outperformed by single nucleotide polymorphisms (SNPs). The lack of a consensus on the genetic distinction between populations has also hindered the development of genetic traceability tools, which rely on the definition of reference units.

While the presence of biologically significant genetic structuring in gilthead sea bream wild populations has been suggested by previous studies the data remain largely equivocal, at least partly due to limitations of the markers used. In the current study a potentially more powerful marker set was employed to document genetic variability and differentiation within this species across North-East Atlantic and Mediterranean basins. Specifically, RAD sequencing was used to simultaneously identify and score a large, genome wide panel of more than 1000 SNPs in a geographically diverse set of sea bream samples (almost 1000 specimens from 23 locations). A second aim of the study was to test the hypothesis that genetic structure is linked to environmental variability. For this, a set of *ad hoc* analyses were performed to characterize the neutral and selected divergence of the populations taking into consideration the effect of environmental variables, as well as demographic processes. The results are discussed in context of the life history of the species and previous knowledge of sea bream's population genetics to

provide a valuable baseline for the future management of wild stocks and for the assessment of potential impacts of aquaculture on the genetics of the species.

## Materials and methods

A total of 956 wild individuals from 23 different locations were sampled (Fig 1 and Table 1). For three sites (GRE-6, GRE-7 and GRE-9) temporal replicates were also available. Samples comprised either fin clips or muscle tissue from dead fish in fish markets. The species is not protected by any of the countries where sampling was performed and all samples came from commercially fished animals; therefore, no specific permission or approval was required for this study. Every effort was made to ensure the fish were likely of wild origin; taken from commercial fishing sources, selecting only larger specimens (i.e. heavier than those usually produced by aquaculture). Tissues were preserved in 95% ethanol as soon as possible after sampling. Genomic DNA was extracted using a commercial column-based kit (Invisorb® Spin Tissue Mini Kit, Invitek, STRATEC Biomedical, 242 Germany) or a salt precipitation method using SSTNE buffer, a modified TNE buffer that includes spermidine and spermine [24], which allowed a more efficient, though more time consuming, extraction of samples that failed with the commercial kit.

Multiple ddRAD libraries were prepared, each including 144 samples, splitting samples from the same population in different libraries in order to avoid confounding library-specific biases. The library preparation protocol followed the original description of Peterson et al. [25], with some modifications that facilitated the efficient screening of a larger number of individuals [26, 27] (see also S1 File).

Raw reads were checked for quality using FASTQC [28]. Then, reads missing a valid restriction site were discarded and barcodes were searched (allowing up to one error) for demultiplexing. Barcodes were removed and the remaining sequences were trimmed to 86 bases in length. Reads with one or more uncalled bases were filtered out, as well as reads with 11 or more consecutive bases with an average quality score less than 20 (1% error rate). If a sample was sequenced on more than one lane, reads were combined into a single file before processing. Stacks 1.3 [29, 30] was used to cluster reads into consensus tags and call high-quality SNPs. Stacks' *de-novo* pipeline was run with minimum depth of coverage to call a stack (-m) set to four; maximum number of differences between stacks to be considered as the same tag

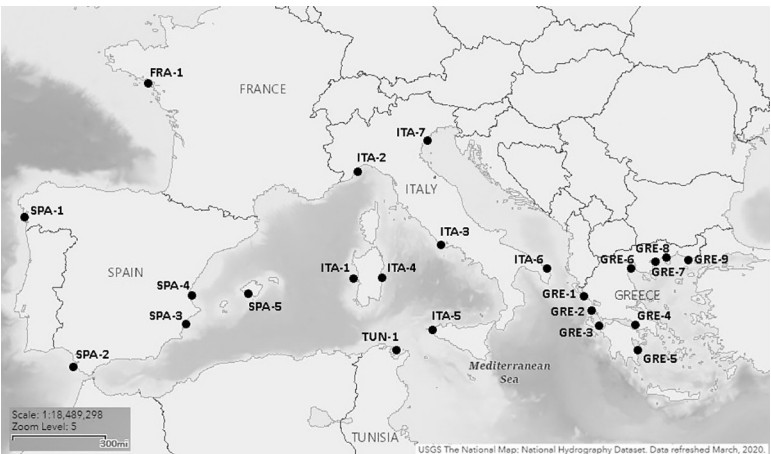

**Fig 1. Map of sampling locations (based on an original image courtesy of the U.S. geological survey).** For more information on sampling locations refer to Table 1.

**Table 1. Sampling groups and basic statistics.**

| Id | Cluster analysis | Location | Latitude | Longitude | Year | No of samples | He | Ho | Fis | Ne (95% upper–lower limits) | Allelic richness | Mean maf at polymorphic loci |
|---|---|---|---|---|---|---|---|---|---|---|---|---|
| FRA-1 | ATL-1 | Noirmoutier | -2.170 | 46.989 | 2003 | 22 | 0.151 | 0.134 | 0.085 | ∞ (4255.9 - ∞) | 1.716 | 0.143 |
| SPA-1 | ATL-2 | Vigo | -8.953 | 41.831 | 2009 | 13 | 0.145 | 0.136 | 0.036 | ∞ (∞—∞) | 1.598 | 0.168 |
| SPA-2 | ATL-3 | Cadiz | -6.400 | 36.500 | 2001 | 17 | 0.151 | 0.142 | 0.044 | ∞ (884.3 - ∞) | 1.672 | 0.154 |
| SPA-3 | WMED-1 | Alicante | -0.317 | 38.286 | 2009 | 20 | 0.143 | 0.124 | 0.082 | ∞ (∞—∞) | 1.654 | 0.149 |
| SPA-4 | WMED-2 | Valencia | -0.100 | 39.500 | 2014 | 24 | 0.154 | 0.138 | 0.073 | ∞ (1821.4 - ∞) | 1.745 | 0.142 |
| SPA-5 | WMED-3 | Balearic | 2.681 | 39.403 | 2013 | 36 | 0.156 | 0.142 | 0.075 | ∞ (9959.9 - ∞) | 1.816 | 0.131 |
| ITA-1 | WMED-4 | W Sardinia | 8.402 | 39.826 | 2002 | 28 | 0.156 | 0.142 | 0.067 | 11201.3 (1418.9 - ∞) | 1.779 | 0.136 |
| ITA-2 | WMED-5 | Genova | 8.901 | 44.360 | 2005 | 33 | 0.153 | 0.133 | 0.092 | 1466.5 (806.1– 7809.2) | 1.779 | 0.136 |
| ITA-3 | WMED-6 | Sabaudia | 12.624 | 41.406 | 2013 | 52 | 0.156 | 0.141 | 0.068 | 1802.2 (1165.4– 3935.5) | 1.870 | 0.122 |
| ITA-4 | WMED-7 | Tortoli | 9.756 | 39.924 | 2002 | 29 | 0.155 | 0.141 | 0.070 | ∞ (∞—∞) | 1.796 | 0.133 |
| ITA-5 | WMED-8 | Trapani | 12.449 | 38.006 | 2007 | 22 | 0.156 | 0.142 | 0.058 | 2606.9 (806.4 - ∞) | 1.739 | 0.145 |
| TUN-1 | WMED-9 | Tunis | 10.602 | 36.932 | 2014 | 106 | 0.154 | 0.133 | 0.105 | 14005.3 (4490.2 - ∞) | 1.931 | 0.112 |
| ITA-6 | WMED-10 | Otranto | 18.532 | 40.360 | 2001 | 20 | 0.154 | 0.146 | 0.033 | 106.4 (94.4–121.7) | 1.686 | 0.155 |
| ITA-7 | ION-1 | Venice | 12.409 | 45.322 | 2014 | 40 | 0.141 | 0.120 | 0.103 | ∞ (6425.9 - ∞) | 1.738 | 0.130 |
| GRE-1 | ION-2 | Ionio | 20.360 | 38.983 | 2014 | 31 | 0.155 | 0.149 | 0.037 | 1660.1 (859.5– 21727.1) | 1.789 | 0.136 |
| GRE-2 | ION-3 | Igoumenitsa | 20.163 | 39.486 | 2006 | 53 | 0.155 | 0.139 | 0.076 | 7102.4 (2242.4 - ∞) | 1.854 | 0.124 |
| GRE-3 | ION-4 | Mesologgi | 21.315 | 38.303 | 2005 | 49 | 0.155 | 0.138 | 0.087 | ∞ (∞—∞) | 1.852 | 0.124 |
| GRE-4 | ION-5 | Korinthiakos | 22.945 | 37.270 | 2013 | 32 | 0.156 | 0.140 | 0.080 | 3157.6 (1173.6 - ∞) | 1.794 | 0.136 |
| GRE-5 | AEG-1 | Nayplio | 22.758 | 38.046 | 2005 | 33 | 0.156 | 0.140 | 0.078 | ∞ (∞—∞) | 1.790 | 0.136 |
| GRE-6b | AEG-2 | Basova Kavalas | 24.495 | 40.846 | 2006 | 49 | 0.155 | 0.137 | 0.091 | 2235.5 (1334.9– 6767.1) | 1.854 | 0.124 |
| GRE-6 | AEG-3 | Basova Kavalas | 24.495 | 40.846 | 2013 | 30 | 0.151 | 0.137 | 0.071 | ∞ (∞—∞) | 1.753 | 0.138 |
| GRE-7b | AEG-4 | Thermaikos gulf | 22.846 | 40.263 | 2004 | 36 | 0.156 | 0.137 | 0.083 | 288.6 (255.6–331) | 1.798 | 0.134 |
| GRE-7 | AEG-5 | Thermaikos gulf | 22.846 | 40.263 | 2013 | 45 | 0.155 | 0.137 | 0.090 | ∞ (9787.2 - ∞) | 1.840 | 0.126 |
| GRE-8 | AEG-6 | Agiasma | 24.419 | 40.644 | 2005 | 43 | 0.157 | 0.140 | 0.071 | 6179 (1991.7 - ∞) | 1.826 | 0.131 |
| GRE-9b | AEG-7 | Alexandroupolis | 25.916 | 40.778 | 2005 | 47 | 0.154 | 0.139 | 0.078 | 29303.4 (2833.1 - ∞) | 1.832 | 0.127 |
| GRE-9 | AEG-8 | Alexandroupolis | 25.916 | 40.778 | 2013 | 46 | 0.155 | 0.138 | 0.087 | ∞ (4241.3 - ∞) | 1.836 | 0.127 |
| Mean | | | | | | 36.8 | 0.153 | 0.138 | 0.074 | - | 1.782 | 0.135 |

*Id* refers to the sampling country; *Cluster analysis* refers to the results of Structure and AMOVA analysis suggesting a subdivision of samples in four main groups (ATLantic, West MEDiterranean Sea, IONian Sea and AEGean Sea); *He*: Expected heterozygosity; *Ho*: Observed heterozygosity; *Fis*: Fixation index; *Ne*: Effective population size; *Mean maf at polymorphic loci*: Average minimum allele frequency at polymorphic loci.

in *ustacks* and *cstacks* (-M and -n, respectively) set to seven; SNP calling model was set to 'bounded'. Correction module *rxstacks* was run after the initial analysis to correct genotypes based on population-wide information (refer to Stacks' website for details about the pipeline). Since including all samples in the catalogue construction would be prohibitively time-consuming with the version of Stacks used, 500 samples were selected for this step, including those with a higher number of reads from each population, in order to have all of them represented. Finally, SNPs were filtered out if scored in less than 80% of the analyzed samples and when

Minor Allele Frequency (MAF) was lower than 0.5%. Similarly, samples were filtered in order to retain only those genotyped at more than 80% of the markers.

Four samples were replicated 12 to 13 times in different libraries in order to assess genotyping precision. For each locus, genotypes of replicates were compared using the most frequent one as a reference and counting the number of mismatches across all replicates.

GenAlEx 6.501 [31] was used to calculate expected (He) and observed (Ho) heterozygosity and allelic richness of polymorphic markers (AR). Deviation from Hardy-Weinberg equilibrium (excess or defect of heterozygotes) was tested for each locus and for each population using Genepop 4.6 [32]. We tested for unusually high LD between pairs of loci using the $r^2$ estimator implemented in Plink 1.9 [33], parsing all pairs of loci. $F_{ST}$ matrices were calculated with Arlequin 3.5.2.1 [34] using 50,000 permutations to test for significance. AMOVA was also performed using Arlequin, to test the clustering suggested by other analyses.

To evaluate the level of genetic diversity of the populations, effective population size (Ne) was estimated using a single sample method based on the increased level of linkage disequilibrium between loci that arises when populations with low Ne are sampled. The algorithm is implemented in NeEstimator 2.01 [35, 36]. Ne were estimated using only putatively neutral SNPs (see below) with minor allele frequency (MAF) >1%. Pairwise genetic relatedness between individuals was calculated using Wang (2002) method as implemented in Coancestry 1.0 [37].

For the clustering and outlier detection analysis described below, when a location was sampled at two different times, the older temporal replicates were excluded from the dataset. Two different approaches were used to summarize and visualize the genetic relationships among groups: the model-based clustering method implemented in Structure [38] and Discriminant Analysis of Principal Components (DAPC) as implemented in the *adegenet* R package [39, 40]. Structure 2.3.4 was run through Parallel Structure [41] to allow faster and more efficient processing using different *k* values and replicates of each *k* value and using the sampling location as *a-priori* information. The analysis was run with *k* ranging from one to ten, each repeated five times to allow evaluation of the likelihood of different simulated number of ancestral clusters. Burn in (BI) was set to 50,000 and the number of iterations (IT) to 100,000. Results from different runs were collated and most likely k values were detected using the Evanno's method implemented in Structure Harvester [42]. A further Structure run was carried out with the most likely *k* value, using 100,000 burn-in cycles and 300,000 iterations. DAPC was carried out using the R package *adegenet* [40]. To avoid the effect of retaining too many principal components (PC), which would describe better the differences between sampled individuals, whilst performing poorly with newly sampled ones, repeated cross-validation was used to select the best number of SNPs and to obtain a trade-off between stability and power of discrimination.

One of the most interesting advantages of genome-wide genotyping is the increased chance of finding genetic regions (i.e. loci) potentially under natural selection. These markers can be used to link genetic and phenotypic traits selected in a particular environment. Bayesian approaches implemented in Bayescan 2.1 [43, 44] and Arlequin was used for this purpose [34]. A combination of different methods is advised to obtain reliable information from the data [45]. Outlier detection analysis was carried out with all samples divided into sampling locations, within Atlantic and Mediterranean basins with samples divided into sampling locations and with samples divided into the four groups suggested by clustering analysis (see Results). Both programs were run with default parameters and finally, an outlier panel (OL) was defined selecting only loci detected by both methods using a stringent threshold (Log10 (PO) > 1.5 for Bayescan and p < 0.05 for Arlequin) to minimize inclusion of false positives. A contrasting neutral dataset was defined as all scored loci excluding the potential outliers suggested by at

least one of the two methods applied. To understand how environmental factors and spatial variables shaped genetic diversity across sea bream distribution range, we applied redundancy analysis (RDA) using R's VEGAN library [46–48]. Environmental data were extracted from the SeaDataNet portal (http://www.seadatanet.org/) which included winter and summer temperature and salinity values at the surface and at 20 m, as these represent functional proxies of more complex environmental variation (S1 Table). Geographic coordinates were referred as close as possible to actual sampling locations for which data were available. The analysis was carried out using the reduced outlier datasets. The significance of relationships between environmental and genetic data was tested with permutation tests with variance explained by different independent factors being taken into account. The panels of explanatory variables were reduced by automatic forward selection based on significant variable criteria and the total proportion of genetic variation was recalculated accordingly. Variation in environmental data values was also compared to variation in allele frequencies to explore potential correlation. using BayEnv 2 [49, 50]. Four separate analyses were run to calculate correlation values and locus/variable specific Bayes Factors (BFs) were averaged to obtain more reliable results. RADtags containing potential outlier loci, as well as loci whose allele frequency was significantly correlated with environmental factors were located in the sea bream genome [51] by BLAST analysis. Genes annotations were extracted from the regions flanking each OL, spanning 50k bp up and downstream from tag location, using a custom Perl script. To detect variation in features under multigenic control, enrichment analysis was carried out for the list of genes extracted with Blast2GO, using the sea bream genome annotation as a reference to identify over or under-represented biological processes, cellular components or molecular functions. Results were reported only for features with p-value < 0.05.

## Results

### Almost one thousand wild sea breams consistently genotyped at 1159 high quality SNP

A total of 767.1 M read pairs were obtained, with an average of 804,463 ± 505,000 reads per individual (range 97–3,104,802 reads). After filtering, an average of 72.2% ± 6% of reads was retained. The initial number of called SNPs was 11,662, included in a total of 216,713 tags. After filtering out low-quality markers, 1165 SNPs (10.6%) were retained. After filtering, the level of data missing per sample ranged from 0% to 19.9%, with an average of 5.4%. Of 30,290 tests for departure from H-W equilibrium carried out, after sequential Bonferroni correction, two loci showed significant deviation from equilibrium (both for an excess of heterozygosity) in more than half of the natural populations and were excluded. A total of 675,703 tests for LD were carried out and four loci pairs showed $r^2$ values higher than 0.7 and for each pair, the locus with lower missing data was retained. The remaining 1159 SNPs were used for subsequent analysis (S2 Table). The mismatch rate between replicated samples at the 1159 filtered loci ranged from 3.4% to 5.8%, with an average of 4.0%. Mean number of alleles per locus (Na) was 1.782, ranging from 1.598 (SPA-1) to 1.931 (TUN-1). Mean expected and observed heterozygosity were 0.153 (range 0.141–0.157) and 0.138 (range 0.120–0.149), respectively (Table 1). All these parameters tended to be lower in the Atlantic populations. Minor Allele Frequency (MAF) ranged from 0.112 to 0.168, with an average of 0.135. Effective Population Size (Ne) for all Atlantic samples were very high ('Infinite'), whereas Mediterranean samples showed values ranging from 106.4 (ITA-6) to 'Infinite' (ten samples). Nevertheless, ranges of confidence varied substantially especially for smaller samples (Table 1), which suggests caution when interpreting these data. The inbreeding index varied from 0.033 to 0.105 and averaged 0.074 (Table 1). No general trend was found for this parameter when comparing Atlantic and

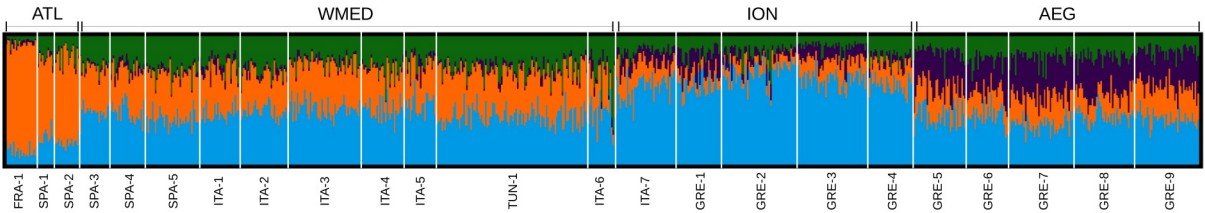

**Fig 2. Structure clustering analysis.** Result for the most likely number of clusters ($k$ = 4). Labels under the graph indicate the sample groups. Labels above the graph indicate the genetic clusters suggested by the analysis and supported by AMOVA.

Mediterranean samples. Relatedness analysis gave clues for the presence of potential full-sib pairs within the SPA-2 sample (Wang relatedness > 0.375). As a consequence, four individuals that were part of highly related pairs were eliminated from the dataset to avoid biases in the clustering analysis. Genetic differentiation between time replicates was not significant. Heterozygosity was also similar for the time replicates, but for Basova Kavalas and Thermaikos Gulf recent samples showed significantly higher Ne (Table 1).

## Four genetic clusters of sea bream populations were identified

Comprehensively, the results obtained with the full SNPs dataset (representing the combination of demographic and selective factors) suggested a subdivision of sea bream into four major genetic clusters: Atlantic (ATL), West Mediterranean (WMED), Ionian/Adriatic seas (ION) and the Aegean Sea (AEG), the strongest differentiation being between Atlantic and Mediterranean samples (Fig 2 and Table 2). The Evanno's method identified four clusters as being most likely and all runs at this number of ancestral groupings showed the same cluster pattern. While samples showed a high level of admixture, those from different basins differed in the proportions of each component. Within-basin differences were much lower (Fig 2). Samples from the Northern and Southern Adriatic Sea (i.e. ION01/ITA7 and WMED10/ITA6, respectively) showed no genetic differentiation ($F_{ST}$ = 0.0000), however under Structure's Bayesian analysis they clustered separately: with ION and WMED populations, respectively. AMOVA analysis further confirmed this subdivision, with the proportion of genetic variability among basins being statistically significant, while the variability within groups was not (Table 2).

DAPC was based on 150 PCs, after cross-validation analysis. Differentiation signal was weak, with the first axis roughly matching the geographical West to the East distribution of the samples. Along the second axis of the DAPC ATL and AEG clusters showed some level of genetic similarity when compared to the two remaining clusters within Mediterranean Sea (S1 Fig).

**Table 2. Results of the AMOVA analysis with sample subdivided according to the clustering analysis.**

| Source of Variation | Degrees of freedom | Sum of squares | Variance components | Fixation indices | Percentage of Variation | P-value |
|---|---|---|---|---|---|---|
| Among groups | 3 | 433.523 | 0.29020 | 0.00363 | 0.36 | <0.01 |
| Among populations within groups | 19 | 732.743 | -0.66080 | -0.00830 | -0.83 | Ns |
| Among individuals within populations | 801 | 68161.210 | 4.82579 | 0.06012 | 6.04 | <0.01 |
| Within individuals | 824 | 62165.500 | 75.44357 | 0.05576 | 94.42 | <0.01 |
| Total | 1647 | 131492.976 | 79.89876 | | - | - |

In general, $F_{ST}$ values were low (average $F_{ST}$ = 0.0031) with a trend of increased values in the comparisons between Atlantic and Mediterranean (S3 Table). Within the Mediterranean, values tended to increase and be more significant in the comparisons between samples from the Western and Eastern Mediterranean (Ionian and Aegean basins). When samples were subdivided into the proposed four groups, $F_{ST}$ values were low but highly significant ($p < 0.001$) and ranging from 0.0015 in the comparison between WMED and AEG to 0.0070 in the comparison between the ION and ATL clusters. No population specific private alleles were observed. However, when analysed at group level, an allele at locus 7704_13 (S2 Table) was observed only within the WMED group, with nine heterozygous individuals being observed (allele freq = 0.024)

## Dissecting genetic differentiation: Atlantic-Mediterranean differentiation at neutral loci and signs of convergent selection at outlier loci

Bayescan analysis detected 20 loci with log10(PO) > 1.5; while Arlequin detected 28 potential diverging outliers at $p < 0.05$. Of these, 16 loci (1.3% of the entire dataset) shared by the two methods were selected to create the 'outlier dataset' (OL, S2 Table). Two additional loci were included in the OL dataset after the analysis with samples divided into four clusters (see above). The OL dataset was subsequently used for analysis focused on exploring the "functional" divergence between populations. The putative neutral loci dataset consisted of 1126 remaining loci, after excluding potential outliers identified by at least one of the approaches used.

Single SNP allele-specific assays were designed and performed for the 16 identified outlier loci set (data not shown). Expected polymorphisms were validated for all but one locus (8727_39), which was removed from subsequent analyses.

BayEnv detected 24 loci, of which only six were among those identified by the joint analysis carried out with Bayescan and Arlequin (S2 Table).

OL allele frequencies did not show a constant behavior when moving from West to East populations (S2 Fig): some followed a gradual variation (e.g. 2689_65); some others showed more abrupt changes at basin boundaries (e.g. 12615_64). finally, OL identified by BayEnv showed patterns of allele frequency that overlapped with the correlated environmental variables variation (e.g. 13310_71).

The genetic structure detected from analysis of the panel of putative neutral loci alone was weaker than that when using the entire dataset, especially within the Mediterranean. Pairwise $F_{ST}$ values were rarely significant, occurring mainly in comparisons between Atlantic samples and samples from Ionian and Aegean seas (S4 Table). Structure clustering analysis, using the most likely value of k = 2 (according to Evanno's method), showed a high degree of admixture, though differentiation was apparent in cluster representations between ATL, WMED and East Mediterranean (ION + AEG) samples (S3 Fig).

The presence of within basin differentiation emerged (from analyses involving the OL panel (S4 Table). The Northern Atlantic sample ATL1/FRA-1 was differentiated from more southern samples ATL2/SPA-1 and ATL3/SPA-2. In general, ATL1/FRA-1 is the most differentiated sample with pairwise $F_{ST}$ values ranging up to 0.1751 (with ION3/GRE-2 sample). WMED was the only group found to be highly homogeneous at outlier loci. Astriking difference uncovered by the OL panel analyses compared to the neutral loci data set was the strong differentiation between Ionian and Aegean samples, reflected in both $F_{ST}$ values and Structure results at the most probable k = 3 (S4 Fig).

Genetic differentiation at outlier loci reflected environmental differentiation between localities, as confirmed by SEASCAPE analysis, which indicated longitude and winter surface

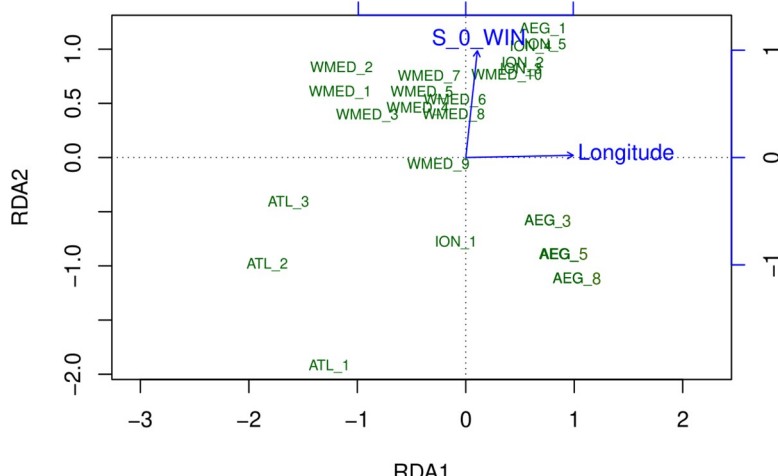

**Fig 3. PCA of the redundancy analysis performed using outlier SNP dataset.** Sampling groups are labeled according to the genetic clusters identified with Structure and AMOVA and plotted according to the most explanatory axes. Longitude and surface winter salinity (S_0_WIN) were the environmental variables most significantly related with genetic data.

salinity (S_0_W) as significant explanatory variables (Fig 3). Climate explained 54% of the variance in the data, while geography explained 14%. The joint effect of the two sets of variables explained 32% of the total variance. The PCA plot indicated that longitude had the strongest effect on the first axis, while winter surface salinity was correlated with the second axis. Along this axis, ATL samples are roughly grouped in the lower half of the graph and closer to AEG and ION-1/ITA-7 samples, reflecting differences in salinity values. ION and ATL samples showed the strongest pairwise differences both in terms of environmental variables and also in terms of differentiation at OL loci.

Almost all of the outlier loci (28 of 34, including those identified by Bayenv) were detected in the genome assembly. Mapped loci were scattered among 16 of the 24 chromosomes of the species. Eighteen of them were located within genes, 11 of which were among the markers indicated by BayEnv as correlated with environmental variables. Fourteen of these tags were located in a Coding DNA Sequence (CDS) region (S2 Table). The enrichment analysis suggested a significant over-representation of genes involved in the biological processes "kinetochore assembly" and "kinetochore organization" (GO:0051382 and GO:0051383 respectively) and "endoplasmic reticulum calcium ion homeostasis" (GO:0032469). In addition, pathways related to the expression of cellular membrane components and many "transferase" molecular functions were highlighted (S5 Table).

## Discussion

While the marine environment appears to be a continuous space with no barrier to gene flow, increasing data support the hypothesis that marine animal populations are structured by oceanographic, historical and other environmental factors. Thanks to the increased power of novel genotyping techniques, the means now exist to detect more subtle, often previously hidden, structure within many marine species. In this study, we employed a genome wide SNP based analysis to more clearly describe the genetic structure of wild gilthead sea bream populations from the Mediterranean Sea and Atlantic Ocean. The genetic differentiation uncovered is likely due to the joint effect of two main factors: (i) a demographic factor derived by long-

term reduced connectivity between populations; and (ii) selection for the environment (i.e. a short-term effect).

## Long term demographic factors explain the neutral genetic structure

$F_{ST}$ levels, calculated from the full SNP dataset, were lower than those usually found in marine fish (average $F_{ST} = 0.062$ [52]) and agree with prior sea bream studies [18, 19]. A previous broad geographical range analysis of gilthead seabream, using allozyme and microsatellite markers (putatively neutral markers) by Alarcón et al. [18] concluded that the population structuring was not associated with known geographic or oceanographic factors. In the current study, a much larger marker panel was employed in an attempt to identify and resolve more subtle genetic structure, if present. Despite pairwise genetic differentiation levels being statistically significant in only a few comparisons, other approaches (i.e. Structure, DAPC and AMOVA) pointed to a significant subdivision of the species in three main basins: the Atlantic, the West Mediterranean and the East Mediterranean, including the Ionian and Aegean seas.

The presence of two major water mass boundaries separating these basins, namely the Almeria-Oran front and the Sicily Channel front, likely explain this pattern. In addition, the Otranto Strait front effect may also explain the differentiation between WMED10/ITA-6 and the Ionian samples. The population structure of another coastal fish species, the East Atlantic peacock wrasse, has previously been associated with these fronts [53]. The authors also found a connection between Ionian and Adriatic samples along the East coast of these basins. The same oceanographic features could explain the similarity between Ionian and North Adriatic (ION1/ITA-7) samples. These patterns could also be influenced by a stronger dispersal along the coast rather than across the opposite stretch of the sea, considering that sea bream prefers shallower water [54].

Evidence of gilthead sea bream genetic differentiation between Atlantic and Mediterranean basins has been previously reported using other typically neutral markers (i.e. microsatellites in [19]), and significant differentiation has also been found between Atlantic samples from north and south coasts of Spain [18]. In the current study, genetic structuring was reflected by loci under selection and thus may be the result of evolutionary responses to different environments (see further below). Atlantic samples collected closer to the Mediterranean (ATL3/SPA-2) were more strongly differentiated from Mediterranean samples than the more distant ATL1/FRA-1 sample. A similar result was found by Alarcón et al. [18], using allozymes and microsatellites. Genetic differentiation within the Mediterranean has been previously reported by other authors [22, 55, 56]. These studies focused on specific areas of the Mediterranean and agreed with our study in detecting the presence of a genetic differentiated cluster in the Adriatic basin [22, 56] and reduced gene flow through Strait of Sicily [55]. They also suggested the presence of differentiation at finer level between samples from West Mediterranean (e.g. between Tyrrhenian and Sardinian seas) based on F-statistics, but this result was not confirmed by their Bayesian analysis [22, 56]. Our results confirm the absence of differentiation, which is not detectable also using the more sensitive OL panel. Low levels of differentiation within basins, coupled with limited differentiation across fronts are not unexpected for this species. Indeed, given the long period of larval dispersal of seabream (>30 days) [57] and its pelagic/benthic vagile lifestyle, retention rates within basins are expected to be low, with fronts and currents having a dominant effect on shaping the neutral genetic structure. In a previous study, based on microsatellite markers [58], some degree of differentiation between samples from the north and south west Mediterranean Sea was reported. This was not apparent in the current study, with the single sample (Tunis) from the south west Mediterranean Sea area grouping with the other West Mediterranean samples.

The lack of previous genetic analyses of sea bream wild populations based on SNPs does not allow for direct comparisons of genetic diversity within the species, but the value reported

here (He = 0.153) was slightly lower than that reported for other species using SNP panels (0.198–0.220 in East Atlantic Peacock wrasse [53]; 0.233–0.290 in European sea bass [59]; 0.270–0.310 in Atlantic herring [60].

## Outlier allele frequency analysis: Convergent adaptation in distant sea bream populations?

Analyses based on outlier SNPs revealed a) a strong subdivision of East Mediterranean wild populations into the two "sub-basins" ION and AEG, and b) similarity between Atlantic and Aegean samples, despite the geographic separation. This is reflected by $F_{ST}$ values showing the highest values between ATL and ION samples and highly significant values also between ION and AEG. Within-basin, significant differences were identified in the Atlantic between the environmentally different groups ATL1/FRA-1 and ATL3/SPA-2, which were not apparent from analysis of the entire SNP dataset. The role of environmental variables is also highlighted by seascape analysis. Separating loci influenced by varying environmental factors from those whose variation is solely due to demographic factors is challenging [61]. The patterns of some OL loci across populations showed allele frequency variations more likely related to geographical distance (e.g. 10524_58), but other OLs seem to reflect the effect of barriers between basins (e.g. 7513_19 and 12615_64). This distinction is important for identifying genes involved in adaptation; in this case, a test based on the correlation between allele frequency and environmental factors is expected to work better [49]. Bayenv analysis added 18 potential outlier loci to those previously detected with Bayescan and Arlequin, many of which showed correspondence with annotated genes and identified two biological processes likely related to adaptation to the environment. While it is acknowledged that the functional analysis conducted was limited, the results point toward a real functional role of the regions sourrounding these SNPs in the adaptation to the environment and deserve further exploration. Therefore, Bayenv, used in combination with redundancy analysis can be useful to detect hidden signature of adaptation despite the reported rate of false positives (around 20–50% [62–64]).

The results presented in this paper are a pivotal step towards the development of practical traceability and conservation tools for the gilthead sea bream. In addition, in a context of massive aquaculture production of this species, an accurate assessment of baseline wild genetic variability and structure is fundamental to the monitoring of potential aquaculture impacts going forward. Sea cage fattening is indeed a source of potential escapees of cultured individuals (often selected for productive traits and/or coming from different areas) into the wild, which could alter the genetic characteristics of wild counterparts [65]. The data from Greek temporal replicates, despite being limited in space, suggest that during the last ten to 15 years the genetic makeup of local wild populations has not been significantly impacted by aquaculture, despite many reported escape events, as also found in other studies of the same area [66].

Assessing population specific neutral and potentially adaptive genetic traits will allow a proper monitoring of the changes that escapees can induce if they successfully reproduce in the wild. The results of our study, based on samples from the entire gilthead sea bream distribution, described the presence of a genetic population structure of the species in the Atlantic and Mediterranean basins and answered biological questions that will support the management of wild stocks of this economically important species and can be used as a baseline for the assessment of the genetic impact of sea cage aquaculture.

## Supporting information

**S1 Fig. DAPC scatterplot of the populations used in this work.** The barchart indicates the Discriminant Axes eigenvalue. Circles represent 95% inertia ellipses.
(PNG)

**S2 Fig. Plot of the allele frequencies of the outlier markers and markers identified as correlated to environmental parameter.** Populations are ordered according to the four genetic clusters identified. For markers identified by Bayenv, values of the correlated environmental parameter are indicated by blue bars.
(PDF)

**S3 Fig. Structure plot for the most likely *k* (= 2) using only the neutral markers.**
(TIF)

**S4 Fig. Structure plot for the most likely *k* (= 3) using only the outlier markers.**
(TIF)

**S1 Table. Table of environmental data used in the seascape analysis and in the correlation analysis of Bayescan.**
(ODT)

**S2 Table. Table of markers retained dafter quality filters and used for the analysis.** SNP: Stacks ID of the marker; Outlier/Bayenv: indicates whether a SNP was identified as an outlier by both Arlequin and Bayescan and/or by Bayenv; Inside gene: indicates if the tag mapped inside a specific gene; CDS: indicates if the tag mapped in the CDS region of the gene.
(XLSX)

**S3 Table. Pairwise $F_{ST}$ values calculated using all the SNPs.** * when p-value < 0.05; ** when p-value <0.01.
(ODT)

**S4 Table. Pairwise $F_{ST}$ values calculated using neutral (above diagonal) and outlier (below diagonal) SNPs.** * when p-value < 0.05; ** when p-value < 0.01.
(ODT)

**S5 Table. Table of the GO terms.** Biological processes, cellular components and molecular functions related to the genes found close to the outlier markers are indicated.
(ODS)

**S1 File. Details of the laboratory protocol for library preparation.**
(DOCX)

## Acknowledgments

The authors would like to thank Giuseppe Muratore of the ASL RM/5- Service of Veterinary Surveillance of the Rome Fish Market [and to APR- Italian Anglers Alliance (part of EAA- European Anglers Alliance)] for providing some of the wild gilthead sea breams samples. The authors would also like to thank the members of the Aquatrace consortium: Danmarks Tekniske Universitet (Denmark): Dorte Bekkevold, Thomas Frank-Gopolos; Università degli Studi di Padova: Rafaella Franch, Serena Ferraresso, Massimiliano Babbucci; Istituto Superiore per la Protezione e la Ricerca Ambientale: Claudia Greco, Donatella Crosetti; Biomolecular Research Genomics: Barbara Simionati, Giorgio Malacrida; Universidade de Santiago de Compostela: Paulino Martinez, Manuel Vera, Miguel Hermida, Fernanda Dotti do Prado; Cluster de la Acuicultura de Galicia: Santiago Cabaleiro, Inés Vila Castro; Katholieke Universiteit Leuven: Filip Volckaert, Gregory Maes, Bart Hellemans; JRC–Joint Research Centre–European Commission: Johann Hofherr, Jann Martinsohn, Marco Ferretti; TRACE–Wildlife Forensic Network Limited: Rob Ogden; Bangor University: Martin Taylor; the University of Stirling:

Michaël Beckaert; Scottish Government–Science and Advice for Scottish Agriculture (SASA): Lucy Webster; Fios genomics Limited: Max Bylesjo; Mustafa Kemal University: Cemal Turan; Institut National de la Recherche Agronomique: Pierrick Haffrey; GIE Laboratoire d'Analyses Genetiques pour les Espaces Animales (France): Lucie Genestout; Ardag Ltd: Glen Pagelson.

## Author Contributions

**Conceptualization:** Francesco Maroso, Konstantinos Gkagkavouzis, Sabina De Innocentiis, Jasmien Hillen, Fernanda do Prado, Nikoleta Karaiskou, John Bernard Taggart, Einar Nielsen, Alexandros Triantafyllidis, Luca Bargelloni.

**Data curation:** Francesco Maroso, Adrian Carr.

**Formal analysis:** Francesco Maroso, Jasmien Hillen, Fernanda do Prado, John Bernard Taggart.

**Funding acquisition:** Einar Nielsen, Luca Bargelloni.

**Investigation:** Francesco Maroso, Einar Nielsen, Alexandros Triantafyllidis.

**Methodology:** Francesco Maroso, Sabina De Innocentiis, John Bernard Taggart, Einar Nielsen, Alexandros Triantafyllidis, Luca Bargelloni.

**Project administration:** Einar Nielsen, Alexandros Triantafyllidis, Luca Bargelloni.

**Software:** Adrian Carr.

**Supervision:** Sabina De Innocentiis, Nikoleta Karaiskou, John Bernard Taggart, Einar Nielsen, Alexandros Triantafyllidis, Luca Bargelloni.

**Validation:** Adrian Carr.

**Visualization:** Fernanda do Prado.

**Writing – original draft:** Francesco Maroso, Konstantinos Gkagkavouzis, Sabina De Innocentiis, Alexandros Triantafyllidis.

**Writing – review & editing:** Francesco Maroso, Konstantinos Gkagkavouzis, Sabina De Innocentiis, Nikoleta Karaiskou, John Bernard Taggart, Alexandros Triantafyllidis, Luca Bargelloni.

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
