## [Decision Letter · Decision Letter 0]

6 Oct 2020

PONE-D-20-20467

Genome-wide analysis clarifies the population genetic structure of wild Gilthead Sea Bream (Sparus aurata)

PLOS ONE

Dear Dr. Maroso,

Thank you for submitting your manuscript to PLOS ONE. After careful consideration, we feel that it has merit but does not fully meet PLOS ONE’s publication criteria as it currently stands. Therefore, we invite you to submit a revised version of the manuscript that addresses the points raised during the review process.

Following the comments raised by the reviewer, introduction and the discussion section should be re written. Particularly in discussion section author should explain the results obtained, comparing these results with previous studies for the other molecular markers. Also note that the reviewer has been especially critical about English grammar of manuscript, which should be improved and edited by a native English speaker

We look forward to receiving your revised manuscript.

Kind regards,

Silvia Martínez-Llorens

Academic Editor

PLOS ONE

Journal Requirements:

2. In your Methods section, please include a comment about the state of the animals following the sampling. Were they euthanized or released? If any animals were sacrificed by the authors, please include the method of euthanasia and describe any efforts that were undertaken to reduce animal suffering.

"The authors have declared that no competing interests exist.".

We note that one or more of the authors are employed by a commercial company: Fios Genomics Ltd.

5.  We note that [Figure 1] in your submission contain [map/satellite] images which may be copyrighted. All PLOS content is published under the Creative Commons Attribution License (CC BY 4.0), which means that the manuscript, images, and Supporting Information files will be freely available online, and any third party is permitted to access, download, copy, distribute, and use these materials in any way, even commercially, with proper attribution. For these reasons, we cannot publish previously copyrighted maps or satellite images created using proprietary data, such as Google software (Google Maps, Street View, and Earth). For more information, see our copyright guidelines: http://journals.plos.org/plosone/s/licenses-and-copyright.

1.     You may seek permission from the original copyright holder of Figure(s) [1] to publish the content specifically under the CC BY 4.0 license.  

Additional Editor Comments (if provided):

Following the comments raised by the reviewer, introduction and the discussion section should be re written. Particularly in discussion section author should explain the results obtained, comparing these results with previous studies for the other molecular markers. Also note that the reviewer has been especially critical about English grammar of manuscript, which should be improved and edited by a native English speaker.

Reviewers' comments:

Reviewer's Responses to Questions

**Comments to the Author**

1. Is the manuscript technically sound, and do the data support the conclusions?

Reviewer #1: Yes

2. Has the statistical analysis been performed appropriately and rigorously? 

Reviewer #1: Yes

3. Have the authors made all data underlying the findings in their manuscript fully available?

Reviewer #1: Yes

4. Is the manuscript presented in an intelligible fashion and written in standard English?

Reviewer #1: No

5. Review Comments to the Author

Reviewer #1: After reviewing the manuscript titled “Genome-wide analysis clarifies the population genetic structure of wild Gilthead Sea Bream (Sparus aurata)”, the authors descript the population genetic structure of S. aurata in the Mediterranean Sea and Atlantic Ocean using RAD-SNP markers. Three and two genetic clusters of this species in the wild populations were detected using neutral markers and outlier markers, respectively. Seascape analysis suggested that this differentiation was mainly due to difference in salinity. The finding is very import to the fishery management of S. aurata. The language is clear. The relevant results of genetic structure analysis are credible. However, I found there are some mistakes as well as concept misunderstanding in this manuscript, despite authors spent tremendous efforts to decipher the population structure of S. aurata with comprehensive tools. This is some problems for the analysis of the data. For the RAD data, it requires a high quality reference genome. If no reference genome, RAD contigs can be assembled as a reference genome. There are several reference genomes for this species in NCBI and ensemble. How can you choose the reference genome for RAD? Please show the detail, how to choose the reference genome for alignment. The second problem, we suggested that RAD method can identify putative loci under selection using Lostin software. The Introduction did not focus on clear background. This section should be rewritten as it describes the aim and methodology of study whereas it should introduce the research topic. In Methodology section, the author should check the other methods for identify putative loci under selection, for examples Lostin analysis. Discussion must be improved removing general statements and explaining really the results obtained and comparing these results with the previous studies for the other molecular markers. I am pleased to inform you that the following paper will not be officially accepted for publication until editing of the English grammer and phrasing by native English speaker. Unfortunately, while I consider some sections of the article very poorly organized in their current form, in my opinion, this manuscript does not meet criteria for publication and must therefore be minor revision. Please find my comments below.

L38-P42, This section will need to be rewritten in order to achieve clarity and give the reader the confidence that the manuscript is going to be worth reading.

P58-L60, The author must mention the results of previous studies in detail.

P66-P72, I suggest that the authors re-write these sentences based on 'hypothesis and test' structure—clear questions raised in the introduction, then based on the question, to collect data and to do relevant analyses rather than every analyses, then in the discussion part, go back to the question defined in the introduction.

L74-L75, The author must mention the sampling information in detail. The author should mention the source of sample. Samples from different genetic backgrounds may has the different results.

L139-L142, We suggested that RAD method can identify putative loci under selection using Lostin software.

L307-L308, The author should mention the more evidence to support this suggestion.

L317-L319, Other studies of? Why the author only cites one reference?

L344-L345, There is the need to add references here to support the argument.

Figure 1 (the map) should have some improved labelling because anyone unfamiliar with the region will have trouble interpreting it. Perhaps the full sample locations could be included as an additional legend in the map.

6. PLOS authors have the option to publish the peer review history of their article (what does this mean?). If published, this will include your full peer review and any attached files.

Reviewer #1: No

---

## [Author Response · Author response to Decision Letter 0]

23 Nov 2020

Dear reviewer,

on behalf of all the authors, thank you for reviewing the manuscript. We are always keen on hearing from our peers about our work and think that constructive criticisms are the best way to improve science. In the revised version that you can find attached, we tried to correct the manuscript as much as possible according to your comments and, where we disagree, we explain analytically our point.

Regarding your general comments, we agree with you that a reference genome can improve the detection of RAD loci. Nevertheless, this depend to a great extent on the quality of the genome itself, with the scoring outcome still reliant on a (different) set of homology parameters. While we acknowledge that the sea bream genome currently available is likely of good quality, we are also confident that the de novo approach taken is valid and does provide a robust set of reliable SNPs for population-based analysis. Many studies have followed such an approach. This approach also gives the (perhaps marginal) advantage of capturing RAD in genomic fragments that are not present in the reference genome. Therefore, we think that only marginal advantages can derive from using a reference-based approach with our dataset.

Regarding outlier detection with other software (e.g. Lositan, as you suggested) please refer to the reply to the specific comment below.

The introduction was improved according to your specific comments. The fact that in some parts it is focused on methodology reflects our intention to stress the fact that new techniques can improve the accuracy of population genetics analysis. This is an important point that we think can make the paper interesting for a wider audience (many might be interested in the sea bream population genetics, but many more are interested in the advantages of using a RAD approach compared to, for example, microsatellites).

Regarding the discussion, and the inclusion of more references to previous works, unfortunately, the body of literature addressing this topic is not very extensive (especially for studies using markers that are not microsatellites), and we think that all the relevant studies regarding gilthead sea bream (and many of related taxa) population genetics were cited. If there is any work that we are not aware of that would substantially improve the discussion, we are open to inclusion if specific publications are suggested. We are aware that using too general sentences is not advisable in this type of works, and we tried to limit these. That said, there are instances where more general remarks can be useful to the reader to view results with a wider perspective.

After one of your main comments, we had the manuscript reviewed by a native English speaker, and we are sure that you’ll find it much better written and clearer.

Please, find below the replies to the specific points in the text.

L38-P42, This section will need to be rewritten in order to achieve clarity and give the reader the confidence that the manuscript is going to be worth reading.

The section was rewritten in order to make it more catchy, but trying to keep the main message of this first part (i.e. marine animals are more tricky to study from a population genetic point of view and therefore researchers need more accurate genotyping techniques, such as that used in our study)

P58-L60, The author must mention the results of previous studies in detail.

We provided more details about the studies mentioned. Please also keep in mind that these works were mentioned again in the discussion, when we compare our results with those from previous studies.

P66-P72, I suggest that the authors re-write these sentences based on 'hypothesis and test' structure—clear questions raised in the introduction, then based on the question, to collect data and to do relevant analyses rather than every analyses, then in the discussion part, go back to the question defined in the introduction.

The last part of the introduction was modified according to the reviewer’s comment. In addition, the discussion was introduced by a sentence that recalls the hypothesis that we wanted to test.

L74-L75, The author must mention the sampling information in detail. The author should mention the source of sample. Samples from different genetic backgrounds may has the different results.

After one of Editor’s comment, we specified that the animals were sampled already dead from local fish markets. We assured that the sampled animals came from the wild by selecting large specimens, considering that farmed sea bream are sold before they reach that size, as far as we are aware.

L139-L142, We suggested that RAD method can identify putative loci under selection using Lostin software.

For this analysis, the most important point for us was to use at least two methods to identify putatively outlier loci. We decided to use Bayescan and Arlequin procedures because they are among the most often cited. In addition, as far as we know Lositan is a Java based software and its usage is getting rarer, as recent operative system don’t allow access to Java programs for security purpose. The addition of a third method (BAYENV) of outlier detection based on environmental variables made us confident that all the genetic variation possibly associated with selection was captured. Therefore we feel that the addition of a fourth statistical package is not needed at this moment

L307-L308, The author should mention the more evidence to support this suggestion.

In the analysis of outlier markers, we found significant correlations between outlier frequencies and environmental variables. These results are discussed in the last part of the discussion. This sentence was modified to invite the readers to look further down in the discussion, where these results are treated more in detail.

L317-L319, Other studies of? Why the author only cites one reference?

Thank you for noting the error. As mentioned before, the studies on this species are not many, and also in this case the work was only one, dated 2009 and using microsatellites. The error was corrected.

L344-L345, There is the need to add references here to support the argument.

What we mentioned in this section is just a hypothesis and we acknowledge that with the results described in this work we cannot go further than that. As reported in the manuscript, for confirming this hypothesis an analysis specifically focused on the outlier loci/genomic regions would be needed but this is beyond the scope of this paper.

Figure 1 (the map) should have some improved labelling because anyone unfamiliar with the region will have trouble interpreting it. Perhaps the full sample locations could be included as an additional legend in the map.

The figure was replaced after Editor’s comments regarding copyright. Figure 1 caption now mentions that more details regarding sampling locations can be found in Table 1. The two will be positioned close to each other in the paper, so we think it will be easy for the reader to have a proper understanding of the sampling map.

Francesco Maroso

---

## [Editor Report · Decision Letter 1]

23 Dec 2020

Genome-wide analysis clarifies the population genetic structure of wild gilthead sea bream (Sparus aurata)

PONE-D-20-20467R1

Dear Dr. Maroso,

We’re pleased to inform you that your manuscript has been judged scientifically suitable for publication and will be formally accepted for publication once it meets all outstanding technical requirements.

Kind regards,

Silvia Martínez-Llorens

Academic Editor

PLOS ONE

---

## [Editor Report · Acceptance letter]

30 Dec 2020

PONE-D-20-20467R1 

Genome-wide analysis clarifies the population genetic structure of wild gilthead sea bream *(Sparus aurata)*

Dear Dr. Maroso:

I'm pleased to inform you that your manuscript has been deemed suitable for publication in PLOS ONE. Congratulations! Your manuscript is now with our production department. 

Kind regards, 

on behalf of

Dr Silvia Martínez-Llorens 

Academic Editor

PLOS ONE